# Counting the Bugs in ChatGPT's Wugs: A Multilingual Investigation into the Morphological Capabilities of a Large Language Model

Leonie Weissweiler[*2,4], Valentin Hofmann[*2-5], Anjali Kantharuban[1], Anna Cai[†1], Ritam Dutt [†1],
Amey Hengle[†6], Anubha Kabra[†1], Atharva Kulkarni[†1], Abhishek Vijayakumar[†1],
Haofei Yu[†1], Hinrich Schütze[2,4], Kemal Oflazer[1], David R. Mortensen[1]

[1]Carnegie Mellon University    [2]LMU Munich    [3]University of Oxford
[4]Munich Center for Machine Learning    [5]Allen Institute for AI    [6]IIT Delhi
weissweiler@cis.lmu.de

## Abstract

Large language models (LLMs) have recently reached an impressive level of linguistic capability, prompting comparisons with human language skills. However, there have been relatively few systematic inquiries into the linguistic capabilities of the latest generation of LLMs, and those studies that do exist (i) ignore the remarkable ability of humans to generalize, (ii) focus only on English, and (iii) investigate syntax or semantics and overlook other capabilities that lie at the heart of human language, like morphology. Here, we close these gaps by conducting the first rigorous analysis of the morphological capabilities of ChatGPT in four typologically varied languages (specifically, English, German, Tamil, and Turkish). We apply a version of Berko's (1958) wug test to ChatGPT, using novel, uncontaminated datasets for the four examined languages. We find that ChatGPT massively underperforms purpose-built systems, particularly in English. Overall, our results—through the lens of morphology—cast a new light on the linguistic capabilities of ChatGPT, suggesting that claims of human-like language skills are premature and misleading.

## 1 Introduction

Do large language models (LLMs) possess human-like linguistic capabilities? With the advent of the latest generation of LLMs such as GPT-4 (OpenAI, 2023b), LLaMA (Touvron et al., 2023), and PaLM (Chowdhery et al., 2022), there appears to be growing evidence for answering this question with *yes* (Bubeck et al., 2023): LLMs are capable of generating text that crowdworkers cannot distinguish from human-generated text (Clark et al., 2021) and excel at linguistic probing tasks such as predicting grammaticality, detecting the subject and tense of

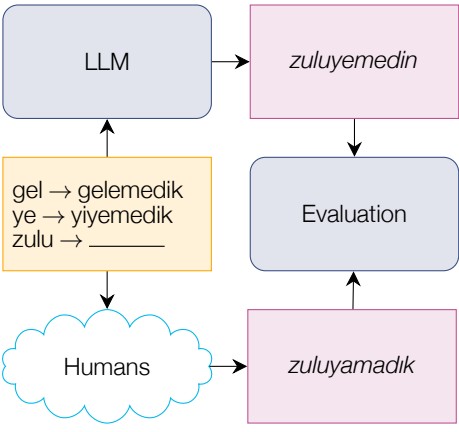

Figure 1: Experimental paradigm for this study (illustrated with Turkish). Human annotators and an LLM are given examples and a nonce word to be inflected. The generated inflected forms are compared.

clauses, and identifying the grammatical number of subjects and objects (Jin et al., 2022).

Despite these encouraging results, the existing body of work has so far examined a relatively limited part of the full spectrum of phenomena that are known to characterize human language, with a heavy focus on syntax and semantics. One area that has been neglected in particular is *morphology*, i.e., the capacity to create words according to systematic patterns of covariation in form and meaning (Haspelmath and Sims, 2010). This gap in the LLM literature is noteworthy given that morphology has been a hallmark of research on computational approaches to language since the very beginnings of neural language processing in the 1980s (Rumelhart and McClelland, 1986b; Plunkett and Juola, 1999; Albright and Hayes, 2002, 2003; Goldberg, 2019).

In this study, we present the first systematic analysis of the morphological capabilities of LLMs, fo-

---

[*]Equal contribution.
[†]Authors sorted alphabetically.

cusing on ChatGPT (OpenAI, 2023a) as the most prominent and most widely-used LLM. Specifically, we investigate ChatGPT's morphological capabilities using the wug test (Berko, 1958), an experimental paradigm in which a participant is asked to provide an inflected or derived form of a nonce word. An example for our evaluation setup is given in Figure 1. Our experiments cover a broad range of morphological constructions and four typologically diverse languages: English, German, Tamil, and Turkish. We find that ChatGPT falls short not only of human performance but also of various supervised baselines.

In sum, our contributions are as follows:

- We conduct the first systematic analysis into the morphological capabilities of LLMs.

- Our study covers a diverse set of morphological constructions/languages and introduces datasets for future research in the area.[1]

- We show that ChatGPT has not achieved human parity—or even state-of-the-art performance—on our nonce-word inflection/reinflection tasks but performs about as well as some older supervised models. We furthermore find evidence for the existence of a real word bias in ChatGPT that is the more pronounced the more data ChatGPT has seen for a given language.

## 2 Related Work

### 2.1 Computational Morphology

Linguists divide morphology into inflection and derivation (Haspelmath and Sims, 2010). While inflection accounts for the different word forms of a lexeme, e.g., *listen*, *listens*, and *listened*, derivation accounts for the different lexemes of a word family, e.g., *listen*, *listener*, and *listenable*. Both inflection and derivation have been addressed in computational linguistics and natural language processing (NLP), albeit with a heavy focus on inflection. One line of work, which is conceptually similar to wug testing, has sought to generate inflected forms, given a stem and a morphological tag (Cotterell et al., 2017a, 2018; Vylomova et al., 2020; Goldman et al., 2022), using systems ranging from weighted finite state transducers and GRU/LSTM encoder-decoder models

(with soft attention or hard monotonic attention) to various transformer models. A special subtype of this task is morphological reinflection, where the input can be a form that is itself inflected (Cotterell et al., 2016a; Kann and Schütze, 2016; Kann et al., 2017; Silfverberg et al., 2017; Pimentel et al., 2021). Other typical tasks in computational research on inflection are morphological segmentation (Cotterell et al., 2015, 2016b,c; Kann et al., 2016), unsupervised morphology induction (Hammarström and Borin, 2011; Soricut and Och, 2015; Xu et al., 2018; Weissweiler et al., 2022), and morphological paradigm completion (Erdmann et al., 2020a,b; Jin et al., 2020). There has also been some interest in the modeling of derivation (Cotterell et al., 2017b; Vylomova et al., 2017; Deutsch et al., 2018; Hofmann et al., 2020b,c).

More recently, there have been a few studies examining the morphological capabilities of language models (Edmiston, 2020; Hofmann et al., 2020a), but they focus on smaller language models such as BERT (Devlin et al., 2019). By contrast, we examine ChatGPT, a model whose parameter count is three orders of magnitude larger, and we analyze its zero-, one-, and few-shot capabilities, an approach fully neglected by prior work.

### 2.2 Multilingual Capabilities of LLMs

Recent studies have extensively examined the evaluation of LLMs in multilingual settings. Some of these studies have specifically investigated the extent to which LLMs can be used for traditional multilingual NLP tasks such as machine translation (Bawden et al., 2022; Hendy et al., 2023; Jiao et al., 2023; Wang et al., 2023). Brown et al. (2023) demonstrate that LLMs perform well across multiple languages even with minimal task-specific training, highlighting their transferability and generalization in multilingual understanding.

### 2.3 LLM Performance on Unseen Data

The fact that LLMs have been pretrained on massive amounts of data means that they have seen and potentially memorized a substantial amount of the items of data used in typical evaluation setups (Magar and Schwartz, 2022). There have been a few attempts in NLP to specifically control for previous exposure (Haley, 2020; Hofmann et al., 2020a; Maudslay and Cotterell, 2021). We follow this idea by generating datasets of novel and uncontaminated nonce words, thus ensuring that the words have not been seen by ChatGPT before.

---

[1]We release our dataset along with our code at `https://github.com/dmort27/chatgpts-wugs`, carefully following the guidelines laid out by Jacovi et al. (2023).

## 3 Data and Morphological Constructions

In this paper, we examine ChatGPT's morphological behavior on a typologically diverse set of languages: English, German, Tamil, and Turkish. While English and German belong to the same language family, German has a more fusional morphological system than English. Turkish is chosen since it is a non-Indo-European language with a fully agglutinative morphology. Tamil is chosen since it is a Dravidian language exhibiting an agglutinative morphology with fusional elements. Thus, in terms of the classical triangle of fusional, isolating, and agglutinative morphologies (Dixon, 1994), the languages cover four different points: almost fully isolating (English), intermediate between isolating and fusional (German), intermediate between fusional and agglutinative (Tamil), and fully agglutinative (Turkish). Furthermore, the chosen languages also cover different points in the spectrum from low-resource to high-resource, enabling us to form hypotheses about the impact of the amount of language-specific training data on the morphological capabilities of an LLM. Statistics for the amount of data in train, dev, and test for the baselines, as well as the number of wug test words, are given in Table 1. We report the accuracy of one annotator at a time against the judgments of all other annotators in Table 2.

### 3.1 English

The English past tense has a long and storied history in computational studies of morphology (Rumelhart and McClelland, 1986a; Pinker and Prince, 1988; Ullman et al., 1997; Plunkett and Juola, 1999; Albright and Hayes, 2002, 2003; Kirov and Cotterell, 2018; Ma and Gao, 2022). English displays a handful of conjugation classes as well as frequent morphographemic alternations—consonant doubling and e-deletion, for example—affecting past forms of verbs.

To create the English data, 50 two- to five-letter irregular verbs (defined as verbs that do not form the past tense simply by adding *-ed*) were sampled from the UniMorph 4.0 dataset (Batsuren et al., 2022). These items were each perturbed by one or two letters (substituting phonetically similar sounds) producing a word not included in UniMorph. These verbs were then annotated by 28 volunteer annotators. Participants were asked to provide the past tense of the nonce word and given an example (*wug → wugged*) and the frame "They

| Lang. | Train | Dev | Test | Wug test |
|---|---|---|---|---|
| English | 10,000 | 1,000 | 1,000 | 50 |
| German | 10,000 | 1,000 | 1,000 | 174 |
| Tamil | 1,541 | 368 | — | 123 |
| Turkish | 8,579 | 851 | 846 | 40 |

Table 1: Data statistics. Please see Appendix A.1 for the distribution of morphological tags across the different splits for the four languages. There was not enough data available for Tamil to form a test set.

| | Accuracy (%) | | |
|---|---|---|---|
| Lang. | @1 | @3 | @5 |
| English | $67.14 \pm 17.76$ | $85.29 \pm 13.06$ | $87.64 \pm 12.13$ |
| German | $63.05 \pm 12.62$ | $83.80 \pm 10.57$ | $87.88 \pm 10.34$ |
| Tamil | $37.09 \pm 26.39$ | $43.85 \pm 26.95$ | $43.85 \pm 26.95$ |

Table 2: Accuracy of one annotator at a time against the judgments of the other annotators on our collected wug dataset, for different values of $k$. For Turkish, since the morphology is deterministic, there is no variation.

{nonce_word} all the time. In fact, they ____ just yesterday." This yielded mappings between a lemma and a ranked list of inflected verbs, e.g., *veed → [veeded, ved, vode]*. The modal annotation was always a regularly inflected form (*-ed* with appropriate allomorphic variation), but other inflectional classes were attested.

### 3.2 German

The German plural of nouns is a morphological phenomenon intensely studied in linguistics and the cognitive sciences due to the general complexity of the alternation between the eight different operations that can be used to express it. German pluralization is particularly notable due to the fact that none of the possible operations express it in a majority of cases (McCurdy et al., 2020). In fact, the most frequent German plural noun suffix *-en* has been argued not to be the default (i.e., the suffix that applies to novel nouns)—an honor that goes to *-s* (Marcus et al., 1995).

To create the dataset of novel German nonce nouns, we drew upon Unipseudo.[2] We generated 200 nonce words with a length between four and seven characters (50 nonce words per character length), using German nouns as input to the algorithm. We then had one German native speaker unrelated to the study (i) generate articles (*der*, *die*, or *das*) for each of the nonce words, and (ii) generate a plural based on the nonce words and the previ-

---

[2] http://www.lexique.org/shiny/unipseudo/

ously selected articles. We manually filtered out words whose plural is blocked by existing German lexemes, resulting in a final set of 174 nonce nouns. These nouns were then annotated by 21 volunteer annotators. Participants were asked to provide the plural of the nonce word and were given an example (*Wug → Wugs*) and the frame "Hier ist ein/e {nonce_word}. Jetzt sind es zwei ____." Similarly to English, this yielded mappings between a lemma and a ranked list of inflected nouns.

### 3.3 Tamil

Tamil is a Dravidian language primarily spoken in regions of South India and Sri Lanka. It is an agglutinative languange in which verbs are conjugated for tense, transitivity, person, number, and (in some cases) gender. For the most part, affixes display allomorphy only due to phonological conditioning and are otherwise invariant across verbs, as is the case with the person/number/gender (PNG) affix (Arden, 1891, 71). This is not the case, however, for tense markers. Among linguists working on Tamil, it is not completely agreed upon how many verb classes there are in the language, with some proposing up to 13 and others as few as three (Lisker, 1951; Agesthialingom, 1971). In the spoken form of Tamil, there are points where verbs are part of completely different classes than their literary counterpart, so in this study we focus exclusively on the written form (Schiffman and Renganathan, 2009).

To simplify the analysis, we utilize a modification of Graul's classification seen in *The English Dictionary of the Tamil Verb*, where there are seven primary classes (Schiffman and Renganathan, 2009). The tense most impacted by these verb classes is the past tense, with each class having a unique form, while the present and future only demonstrate three forms across the classes. As such, we focus on the past tense and designate the same transitivity (intransitive) and PNG (third person singular masculine) affix across all experiments. In examining this, we gain information about the ways LLMs handle morphologically complex languages with inflectional classes defined in both phonological and morphological terms. This contrasts with English, where inflection is not agglutinative, and Turkish, where morphology is agglutinative but where there are no inflectional classes.

To create a dataset for training the baseline models and generating samples for the few-shot

| Features | Example |
|---|---|
| First person singular agreement and past tense | zöbür-ür-üm → zöbür-dü-m |
| Second person plural agreement, reported/inferential past tense, and negative polarity | zöbür-ür-sünüz → zöbür-me-miş-siniz |
| Dative case, first person possessive | zürp-ten → zürb-üm-e |
| Accusative singular | börüt → börüd-ü |

Table 3: Turkish tasks. Forms with colored suffixes are actually used in the long prompt in a contextually meaningful short sentence. Hyphens represent morpheme boundaries. The last row is for simple inflection. The predicted forms (to be predicted, on the right) have the following morphosemantics: "I [verb]-ed", "(I heard that) you have not [verb]-ed", "to my [noun]", "the [noun] (as a definite object)".

prompts, 86 common Tamil verbs were sampled and conjugated with every possible combination of tense and PNG suffixes. These conjugations were generated automatically and then validated by two native speakers for accuracy. Unlike in the nonce word case, there was 100% agreement between speakers. The nonce words were generated by combining syllables from real verb roots and checking against a Tamil dictionary to assure the words created were not real. Nonce verbs were created to be between two and six letters long to best match the distribution of real Tamil verbs. In order to get the "correct" past tense for these verbs, five native Tamil speakers were asked to provide past tense forms (e.g., நிடு *niṭu* → [நிடுத்தான் *niṭuṯ:a:n*, நிட்டான் *niṭ:a:n*, நீடினான் *ni:ṭina:n*]). The mode of these responses was taken to be the gold form, with the level of agreement amongst speakers recorded for later analysis. The comparatively lower inter-annotator agreement can be explained by the lack of historical and linguistic context given to the annotators, since a large part of classification is historical.

### 3.4 Turkish

Turkish is an agglutinative language where words consist of multiple morphemes attached to a root. Surface realizations of morphemes are influenced by deterministic morphophonological processes like vowel harmony, consonant assimilation, and elision. Unlike many other languages, Turkish has complex word form morphotactics, particu-

larly when multiple derivations are present.

To simplify the task and reduce the number of feature combinations, we utilized four datasets with different levels of complexity and a limited number of inflectional features. In most cases, the context provides an inflected form with one set of features, and the model must predict the form with the requested set of features. The first three tasks are reinflection tasks, demanding proficiency in both morphotactics and morphographemics. The fourth task is a straightforward inflection task (see Table 3). Each task consists of up to five shot examples for real roots and 10 test examples with nonce roots. Stimuli and gold annotations were produced by our (single) Turkish annotator.

## 4  Methodology

We compare the outputs of ChatGPT under a variety of prompting regimens and a substantial set of supervised baselines (both neural and non-neural) to human annotations of the data described in Appendix 3. Results are evaluated using accuracy at $k$ ($acc@k$), i.e., a model's response is regarded as correct if it is in line with any of the top $k$ human responses. This evaluation method takes into account inter-speaker morphological variability, which is more wide-spread than previously thought (Dammel and Schallert, 2019).

### 4.1  Baselines

We investigate the efficacy of several baselines for the task of morphological inflection. The chosen baselines encompass both statistical and neural architectures that have shown impressive performance on the morphological generalization task in recent years. We evaluate their performance on the SIGMORPHON 2023 task as well as on our constructed wug test set. The baselines have complementary strengths (see Section 5).

#### 4.1.1  Training Data

We used the train/dev/test splits of the SIGMORPHON 2023 Inflection Shared Task[3] for English and German. The choice of the train/dev/test splits was motivated by the fact that there was no overlap of lemmata between the individual splits, thus mimicking a wug-like setting.

The Turkish training data for baselines was generated directly using a Turkish morphological ana-

lyzer/generator (Oflazer, 1994), because the aforementioned SIGMORPHON 2023 dataset did not have a sufficient number of examples for most of the feature combinations. The morphological generator was set up to generate only Turkish word forms that corresponded to the selected inflectional morpheme combinations we selected, for *all* applicable roots. For testing, we expected the baseline systems to generate the word forms with the selected inflectional feature combinations, but *for 10 nonce roots*. The nonce roots were chosen so that they would force the inflected forms to orthogonally adhere to surface morphographemic constraints and rules such as various types of vowel harmony, consonant elision, or assimilation at morpheme boundaries.

Similarly, for Tamil, we split the data into train and dev sets. Since we have a limited amount of Tamil data, we kept the split ratio at around 4:1 between train and dev sets.

We report the results of all baselines in Table 4. Baselines generally perform as expected, validating our usage of them. It should be noted that MinGen and AED are evaluated in IPA/feature space and may therefore be at a disadvantage compared to baselines operating directly in orthography. The training data was converted from orthography into IPA using Epitran (Mortensen et al., 2018).

#### 4.1.2  Affix Rule Learner (ARL)

As a baseline for the 2020 and 2021 SIGMORPHON shared tasks, a simple non-neural system (Liu and Mao, 2016) was implemented that uses edit distance to "discover prefix and suffix rules in training data."[4] At test time, the system modifies a lemma by applying the longest matching suffix rule and most frequently applied prefix rule for a given morphosyntactic description.

#### 4.1.3  Minimal Generalization Learner (MinGen)

Wilson and Li (2021) proposed a minimal generalization model based on a simplified form of Albright and Hayes (2002) to learn morphological rules. First, base rules that describe the changes needed to convert a lemma to an inflected form are generated from training data. The rules are further generalized by comparing phonological features of the rule contexts. The rules are then scored by a confidence metric based on their accuracy and

---

[3] https://github.com/sigmorphon/
2023InflectionST

[4] https://github.com/sigmorphon/2021Task0/tree/
main/baselines

| | English | | German | | Turkish | | Tamil |
|---|---|---|---|---|---|---|---|
| Model | Dev | Test | Dev | Test | Dev | Test | Dev |
| ARL | 95.40 | 96.60 | 77.40 | 79.80 | 94.36 | 93.50 | 85.60 |
| MinGen | 81.40 | 78.70 | 72.70 | 70.70 | 93.65 | 93.03 | 87.23 |
| FIT | $96.22 \pm 0.19$ | $94.93 \pm 0.49$ | $79.01 \pm 1.16$ | $81.04 \pm 1.39$ | $97.00 \pm 0.22$ | $96.25 \pm 0.26$ | $64.24 \pm 3.11$ |
| PPI | $95.95 \pm 0.63$ | $94.74 \pm 0.90$ | $73.57 \pm 5.37$ | $78.26 \pm 4.66$ | $96.61 \pm 0.60$ | $96.56 \pm 0.66$ | $76.76 \pm 2.10$ |
| AED | $71.06 \pm 5.74$ | $70.16 \pm 5.79$ | $64.44 \pm 1.85$ | $67.44 \pm 2.02$ | $95.54 \pm 0.77$ | $95.19 \pm 1.41$ | $50.70 \pm 2.84$ |

Table 4: Results (*acc@k*) of the baselines on our development and test data. See Section 4.1.1 for full details.

scope. At test time, the rule with the highest score among the applicable rules is used.

### 4.1.4 Feature Invariant Transformer (FIT)

Wu et al. (2021) proposed a simple technique employing a character-level transformer for feature-guided transduction that was used as a baseline for the 2021 SIGMORPHON shared task.[5] This is a generative model capable of performing character-level decoding to generate target inflections. In comparison to a vanilla transformer model, positional counts are used only for characters and not for features. The model also incorporates unique tokens to mark whether a given token is a feature.

### 4.1.5 Principle Parts for Inflection (PPI)

We apply the approach of Liu and Hulden (2020), which recasts the task of morphological inflection as a "paradigm cell filling problem." This leverages a lexeme's principal parts—the minimum subset of paradigm slots needed to generate the other slots in its paradigm. Specifically, for low-resource scenarios, the principal parts of a paradigm identify additional slots that are crucial in generating the target-inflected lemma.

### 4.1.6 Analogical Encoder-Decoder (AED)

Following up on Albright and Hayes (2003) and Kirov and Cotterell (2018), Calderone et al. (2021) proposed a recurrent neural network encoder-decoder architecture augmented with pre-compiled analogical patterns for generating morphological inflections of nonce words. This model leverages the UniMorph Tags and fine alternation pattern (FAP) associated with each lemma in relation to its inflection form. FAPs analyze the positioning of word forms within the system to identify recurrent patterns representing conventional linguistic elements.

[5] https://github.com/sigmorphon/2021Task0/tree/main/baselines

### 4.2 Prompting

We employ three distinct prompting styles, namely zero-, one-, and few-shot, to interact with the language model. We start with a simple instruction in each language, for example:

> "Fill in the blank with the correct past tense of the word 'wug'. Give your response in one word.
> They wug all the time. In fact, they ____ just yesterday."

For Tamil, the instruction portion of the prompt is omitted because of ChatGPT's unreliable performance when given instructions in that language. We select one example with real words for each major inflection class of the phenomenon in question. We then perform multiple runs: 10 for the zero-shot scenario, one for every shot for the one-shot scenario, and 10 for the few-shot scenario, with a new random permutation of all examples each time. We query `gpt-3.5-turbo-0613`, select the first word of the response, and filter by removing non-word characters. We evaluate by computing the accuracy for each of the runs, averaged over all queried nonce words, and compute the mean and standard deviation across all runs. We employ *acc@k* as our evaluation metric, setting $k = 5$ for our main evaluation. We provide results for $k = 1$ and $k = 3$ in Appendix A.4. The $k$ gold forms are the $k$ responses most frequently generated by humans. Since only one Turkish response is possible (the morphology is deterministic), $k$ is always 1 for this language. We then perform an additional experiment for comparison in which we remove the context around the nonce word and only give the instructions as well as the last line. We call this the *short* prompt and the original described above the *long* prompt. We provide instances of *long* and *short* prompt in Appendix A.5.

## 5 Results

### 5.1 Overall Performance

For *acc@*5, the performance of ChatGPT never exceeded that of the strongest baselines (ARL, AED, and PPI) regardless of the prompting regime, as shown in Table 5. However, it beats certain older baselines such as MinGen (the minimum generalization learner). ChatGPT performed best when it was explicitly prompted to complete an analogy with a single example (i.e., short 1-shot), as can be seen in Figure 2. We observe that similar trends hold for *acc@*1 and *acc@*3 (see Appendix A.4), but the gap between the strongest baselines and ChatGPT decreases with *k*.

**English** ChatGPT's performance on English was uniformly worse than both the average annotator (87.64%) and the strongest baselines. *acc@*1 falls below 60% in the 0-shot condition but is markedly better when shots are supplied. Short prompts, which require the model to complete a simple analogy, resulted in better performance than long prompts. In all conditions, authentic English words that did not occur in the reference annotations appeared as outputs when the nonce word and the authentic word were orthographically similar (see the discussion in Section 6.4).

**German** The best German result was 88.94% (short 1-shot), which beat all of the baselines except for ARL and FIT. The other results are similarly strong in contrast to the other languages. The impact of *k* is not noticeable here. This, in combination with the fact that the human performance on *acc@*5 was 88%, indicates that the task is perfectly performed by ChatGPT. It has reached the upper bound given by the inherent subjectivity of the task (reflected in the human variability) and the impact of *k* is, therefore, not measurable. This is further solidified by the very small impact of the long vs. short prompts.

**Tamil** Tamil performance of ChatGPT was significantly worse than the provided baselines, even in the few-shot conditions. For the few-shot case, there was marginally better performance when using short prompts, but this did not apply to the 0- or 1-shot case (in which no accurate outputs were generated). Across the board, the performance on Tamil was markedly worse than performance on English and German. However, considering that the average annotator had only 43.85% accuracy

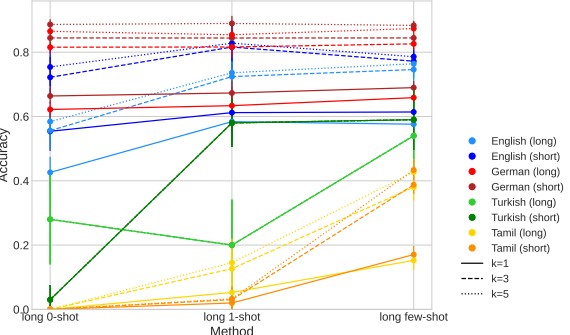

Figure 2: Results for the different prompt scenarios, formats, languages, and values of *k*.

against the judgments of the other annotators, the few-shot accuracy is quite reasonable.

**Turkish** The prompting performance for the Turkish inflection task is worse than for English and German, especially in the long prompt case. For this task, the morphotactics is trivial but the selection of the allomorph depends on stem vowels, stem-final consonants, whether there is a consonant cluster ending the stem, and whether the stem is monosyllabic or not. ChatGPT gets better results with the short prompt through an analogical example. For the three reinflection tasks, ChatGPT gets mixed results that are overall worse than for the inflection task (see Table 6).

## 6 Analysis

### 6.1 The Nature of the Task

The inherent complexity of the inflection tasks for the various languages (and the reinflection task for Turkish) varies greatly. English and Turkish are the simplest: the top-ranked form can always be obtained by adding a single suffix and applying a few morphographemic alternations. German annotations show no dominant pattern and assign nonce words to morphological classes according to complex criteria. However, German performance is clearly better, suggesting that factors other than inherent complexity play a role in ChatGPT's ability to generalize morphological patterns.

### 6.2 Impact of Tokenization

There is mounting evidence that the morphologically suboptimal nature of many tokenizers may limit the morphological capabilities of LLMs (Bostrom and Durrett, 2020; Hofmann et al., 2021). ChatGPT's tokenization, i.e., byte-pair encoding

| Method | English | German | Tamil | Turkish |
|---|---|---|---|---|
| ARL | 100.00 | 94.25 | 61.48 | 60.00 |
| MinGen | 62.00 | 64.37 | 49.18 | 40.00 |
| FIT | 98.00 ± 1.26 | 92.87 ± 0.74 | 63.28 ± 3.36 | 67.00 ± 4.58 |
| PPI | 94.60 ± 2.54 | 85.98 ± 5.91 | 55.33 ± 1.84 | 68.00 ± 4.00 |
| AED | 57.60 ± 6.62 | 48.51 ± 5.45 | 58.69 ± 5.46 | 56.00 ± 4.90 |
| long 0-shot | 58.40 ± 5.28 | 86.49 ± 1.07 | 0.00 | 28.00 ± 14.00 |
| long 1-shot | 73.60 ± 6.97 | 85.42 ± 2.52 | 14.52 ± 7.48 | 20.00 ± 14.14 |
| long few-shot | 76.40 ± 4.45 | 87.36 ± 2.37 | 42.70 ± 3.96 | 54.00 ± 10.20 |
| short 0-shot | 75.40 ± 5.87 | 88.62 ± 1.64 | 0.00 | 3.00 ± 4.58 |
| short 1-shot | 82.80 ± 5.60 | 88.94 ± 2.35 | 3.28 ± 3.99 | 58.00 ± 7.48 |
| short few-shot | 78.60 ± 2.84 | 88.33 ± 1.15 | 43.36 ± 3.12 | 59.00 ± 9.43 |

Table 5: Results ($acc@k$) for all languages ($k = 5$ except for Turkish where $k = 1$, cf. Section 4.2).

| Type | 0-shot | 1-shot | few-shot |
|---|---|---|---|
| long | 3.00 ± 1.80 | 20.67 ± 5.73 | 33.33 ± 4.94 |
| short | 7.00 ± 4.33 | 18.67 ± 6.18 | 31.00 ± 4.23 |

Table 6: Results for Turkish averaged over the three reinflection tasks ($k = 1$).

(Sennrich et al., 2016), has been shown to be particularly problematic (Bostrom and Durrett, 2020; Hofmann et al., 2022).

To examine the impact of tokenization, we measured the number of tokens into which the nonce words are split for the individual languages and computed the accuracy as a function of the number of tokens. Our hypothesis was that longer token sequences are less optimal, potentially leading to worse performance. However, using two-sided $t$-tests, we did not find a significant difference between nonce words with different token lengths. We interpret this as indicating that tokenization plays a less pronounced role for ChatGPT.

### 6.3 Impact of $k$

We observe that the gap between the baselines and our results increases with $k$ (see Table 5, Appendix A.4), suggesting that ChatGPT tends to generate either a top-ranked form or an implausible inflection while the baselines tend to produce plausible inflections which are less frequent in the human annotations. ChatGPT's penchant for implausible inflections may be a result of its real word bias (see Section 6.4 below).

### 6.4 Real Word Bias

In English and German—and to a lesser extent in Turkish—many of the forms generated by Chat-GPT belong to a different lexeme than the nonce word and thus do not constitute inflections in any strict linguistic sense (see Section 2.1). Crucially, the stem of the generated form is always a real word (i.e., a word that exists in the respective language). Examples of this phenomenon include, for English: *did* as the past tense of *dedo*, *blushed* as the past tense of *blus*, *fried* as the past tense of *fride*; and for German: *Ozeane* ('oceans') as the plural of *Ozeak*, *Institute* ('institutes') as the plural of *Instite*, *Sklaven* ('slaves') as the plural of *Schlave*. It is important to notice that in all these cases, (i) the generated form has the correct morphological properties—e.g., the English forms *did*, *blushed*, *fried* are indeed past tense forms—but the stem is a real word rather than the nonce word, and (ii) the stem that is generated in lieu of the nonce word is a frequently occurring word in the respective language and has a certain (sometimes strong) orthographic similarity to the nonce word. We denote this tendency *real word bias*.

The concept of real word bias allows us to make a hypothesis about the way in which ChatGPT addresses morphological tasks. We think ChatGPT is *not* applying morphological rules to a stem, which would be in line with item-and-process accounts of morphology (Hockett, 1954). Rather, it seems to linguistically decode the point in its representational space defined by the semantic constraints in the prompt. In cases where this point (and its immediate neighborhood) is unoccupied, it generates a form based on the nonce word, but in cases where there is a form of a real word close to the point (e.g., because of superficial orthographic similarity), it generates this form instead. The fact that the real word bias is strongest for German and English (the two high-resource languages) suggests that the representational space is more dense for these two languages, increasing the probability that there is a real word close to the point that the model is trying

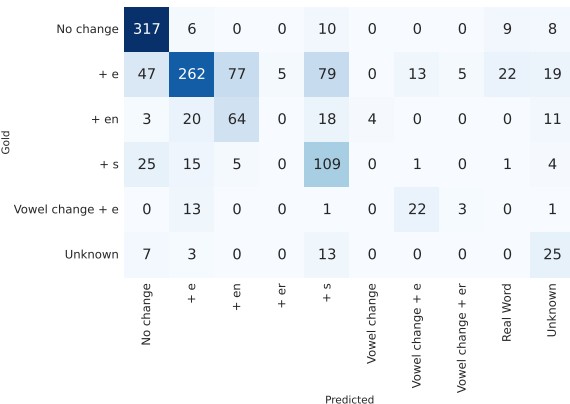

Figure 3: Confusion matrix for competing German plural morphemes for the few-shot setting.

to decode based on the prompt.

### 6.5 Morphological Productivity

The productivity of a morpheme is traditionally defined as its propensity to be used in novel combinations (Plag, 1999; Bauer, 2001; Haspelmath and Sims, 2010). Crucially, morphemes with the same meaning can differ in their productivity—for example, for English deadjectival nominalizing suffixes, *-ness* (e.g., *robustness*) is generally more productive than *-ity* (e.g, *equality*), which in turn is more productive than the fully non-productive *-th* (e.g., *warmth*). We are interested to see whether there is any difference in the productivity of morphological patterns exhibited by ChatGPT compared to the human sample. We focus on German as it has the most complex pattern of competing morphemes, and we examine the few-shot results as they show the best performance overall.

We start by comparing the distribution over alternative plural morphemes generated by ChatGPT with the human responses. As shown in Figure 3, there are several morphemes that are used by Chat-GPT similarly to humans (e.g., the null morpheme). Cases of overgeneralization, where ChatGPT systematically generalizes the usage of a particular suffix to contexts where the suffix is not used by humans, are mainly limited to two plural morphemes: *-en* (77 generations for gold morpheme *-e*) and *-s* (79 generations for gold morpheme *-e*). Interestingly, these two plural morphemes are the two most productive plural morphemes in German (Köpcke, 1988). This indicates two important points: (i) ChatGPT is sensitive to the productivity of morphemes, i.e., it has acquired the ability to model how productive certain morphemes are

as a result of pretraining; (ii) it does not identically mirror the behavior of humans, but rather amplifies the productivity of certain morphemes. The finding that the most productive morphemes (for humans) are becoming more productive for Chat-GPT while the least productive morphemes (for humans) are becoming less productive for ChatGPT bears some theoretical resemblance to discussions about bias amplification (Ahn et al., 2022).

## 7 Future Directions

Morphological patterns are only one kind of generalization that can be investigated through a wug-like experimental paradigm. The form-meaning relationships encoded in language and multimodal models, including constructional and iconic pairings, can be investigated through prompting with nonce stimuli, leading to new insights regarding the generalizations they capture.

### Limitations

Our research was conducted with a single model (gpt-3.5-turbo-0613), so it is not certain that our results will generalize to other versions of GPT-3 or to GPT-4, let alone other LLMs. Although we went to great lengths to develop prompts that would maximize ChatGPT's performance on the tasks, it is not possible to state definitively that another strategy would not produce better performance. While the languages were typologically varied, it is not clear whether the results observed in the current study are generally robust or are coincidental properties of the small set of languages and datasets under investigation. Furthermore, comparing the languages to one another is problematic because it was not possible to control other variables while varying the language. For example, the English and Tamil tasks involve verbal inflection while the German and Turkish tasks involve nominal inflection. Finally, the number of annotators for Tamil was very small and inter-annotator agreement was very low, meaning that the results of the Tamil experiments must be approached with special caution (but see our discussion about morphological variation in Section 3).

### Ethics

LLMs are already impacting the world's people in significant ways, for good and ill. Understanding their limitations, particularly with regard to non-hegemonic language communities, is an ethical im-

perative. This study highlights one specific way in which an LLM should not be treated as a surrogate human, thus motivating additional research on language modeling for structurally diverse and low-resource languages.

## Acknowledgements

This work was funded by the European Research Council (#740516). The second author was also supported by the German Academic Scholarship Foundation. We thank the reviewers for their extremely helpful comments.

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

## A Appendices

### A.1 Morphological Tags

In Table 7, we provide details about the morphological tags that are comprised by the train, dev, test, and wug test sets for the four languages. The tags for English (eng), German (deu), and Tamil (tam) are defined in accordance with the description in UniMorph 4.0 dataset. For Turkish (tur),the tags are defined in Section 3.

### A.2 Hyperparameter Tuning

For all baselines, we follow the hyperparameter settings from the publicly available code repositories. The only exception is AED, where the number of epochs was increased from 40 to 200.

### A.3 Qualtrics Details

Our study leveraged Qualtrics, a robust and comprehensive survey software tool that facilitates the

| Lang | Tags | Train | Dev | Test | Wug |
|---|---|---|---|---|---|
| eng | V;NFIN | 2015 | 206 | 202 | 0 |
| eng | V;PRS;NOM(3,SG) | 1987 | 190 | 213 | 0 |
| eng | V;PST | 1981 | 198 | 185 | 50 |
| eng | V;V.PTCP;PRS | 2018 | 201 | 200 | 0 |
| eng | V;V.PTCP;PST | 1999 | 205 | 200 | 0 |
| deu | V.PTCP;PRS | 246 | 19 | 19 | 0 |
| deu | V;IMP;NOM(2,PL) | 246 | 19 | 16 | 0 |
| deu | V;IMP;NOM(2,SG) | 241 | 22 | 17 | 0 |
| deu | V;IND;PRS;NOM(1,PL) | 246 | 20 | 18 | 0 |
| deu | V;IND;PRS;NOM(1,SG) | 227 | 21 | 17 | 0 |
| deu | V;IND;PRS;NOM(2,PL) | 250 | 29 | 21 | 0 |
| deu | V;IND;PRS;NOM(2,SG) | 258 | 25 | 15 | 0 |
| deu | V;IND;PRS;NOM(3,SG) | 233 | 21 | 22 | 0 |
| deu | V;IND;PST;NOM(1,PL) | 235 | 26 | 28 | 0 |
| deu | V;IND;PST;NOM(1,SG) | 236 | 17 | 11 | 0 |
| deu | V;IND;PST;NOM(2,PL) | 257 | 20 | 23 | 0 |
| deu | V;IND;PST;NOM(3,PL) | 243 | 15 | 29 | 0 |
| deu | V;IND;PST;NOM(3,SG) | 247 | 22 | 27 | 0 |
| deu | V;NFIN | 248 | 21 | 13 | 0 |
| deu | V;SBJV;PRS;NOM(1,PL) | 234 | 25 | 18 | 0 |
| deu | V;SBJV;PST;NOM(1,PL) | 243 | 20 | 20 | 0 |
| deu | V;SBJV;PST;NOM(2,SG) | 229 | 27 | 24 | 0 |
| deu | V;SBJV;PST;NOM(3,PL) | 247 | 22 | 22 | 0 |
| deu | N;ACC(PL) | 368 | 44 | 49 | 0 |
| deu | N;ACC(SG) | 385 | 47 | 53 | 0 |
| deu | N;DAT(PL) | 361 | 44 | 48 | 0 |
| deu | N;DAT(SG) | 382 | 47 | 52 | 0 |
| deu | N;GEN(PL) | 364 | 44 | 48 | 0 |
| deu | N;GEN(SG) | 385 | 47 | 50 | 0 |
| deu | N;NOM(PL) | 370 | 44 | 49 | 174 |
| deu | N;NOM(SG) | 391 | 47 | 53 | 0 |
| deu | V.PTCP;PST | 242 | 23 | 23 | 0 |
| deu | V;IND;PRS;NOM(3,PL) | 232 | 25 | 19 | 0 |
| deu | V;IND;PST;NOM(2,SG) | 215 | 25 | 18 | 0 |
| deu | V;SBJV;PRS;NOM(2,PL) | 248 | 20 | 23 | 0 |
| deu | V;SBJV;PRS;NOM(3,PL) | 247 | 26 | 26 | 0 |
| deu | V;SBJV;PRS;NOM(3,SG) | 238 | 26 | 24 | 0 |
| deu | V;SBJV;PST;NOM(3,SG) | 261 | 22 | 26 | 0 |
| deu | V;SBJV;PRS;NOM(1,SG) | 246 | 22 | 16 | 0 |
| deu | V;SBJV;PST;NOM(1,SG) | 238 | 21 | 28 | 0 |
| deu | V;SBJV;PST;NOM(2,PL) | 239 | 17 | 17 | 0 |
| deu | V;SBJV;PRS;NOM(2,SG) | 222 | 18 | 18 | 0 |
| tur | V;POS;PAST;A1SG | 2005 | 201 | 202 | 10 |
| tur | V;NEG;NARR;A2PL | 2005 | 201 | 202 | 10 |
| tur | N;A3SG;P1SG;DAT | 2170 | 214 | 214 | 10 |
| tur | N;A3SG;PNON;ACC | 2172 | 214 | 214 | 10 |
| tam | V;PRS-1SG | 67 | 16 | 0 | 0 |
| tam | V;FUT-1SG | 67 | 16 | 0 | 0 |
| tam | V;PST-2SG | 67 | 16 | 0 | 0 |
| tam | V;PRS-2SG | 67 | 16 | 0 | 0 |
| tam | V;FUT-2SG | 67 | 16 | 0 | 0 |
| tam | V;PST-3SG.M | 67 | 16 | 0 | 123 |
| tam | V;PRS-3SG.M | 67 | 16 | 0 | 0 |
| tam | V;FUT-3SG.M | 67 | 16 | 0 | 0 |
| tam | V;PST-3SG.F | 67 | 16 | 0 | 0 |
| tam | V;PRS-3SG.F | 67 | 16 | 0 | 0 |
| tam | V;FUT-3SG.F | 67 | 16 | 0 | 0 |
| tam | V;PST-3SG.HON | 67 | 16 | 0 | 0 |
| tam | V;PRS-3SG.HON | 67 | 16 | 0 | 0 |
| tam | V;FUT-3SG.HON | 67 | 16 | 0 | 0 |
| tam | V;PST-1PL | 67 | 16 | 0 | 0 |
| tam | V;PRS-1PL | 67 | 16 | 0 | 0 |
| tam | V;FUT-1PL | 67 | 16 | 0 | 0 |
| tam | V;PST-2PL | 67 | 16 | 0 | 0 |
| tam | V;PRS-2PL | 67 | 16 | 0 | 0 |
| tam | V;FUT-2PL | 67 | 16 | 0 | 0 |
| tam | V;PST-3PL | 67 | 16 | 0 | 0 |
| tam | V;PRS-3PL | 67 | 16 | 0 | 0 |
| tam | V;FUT-3PL | 67 | 16 | 0 | 0 |

Table 7: Distribution of tags over the different splits for the four languages.

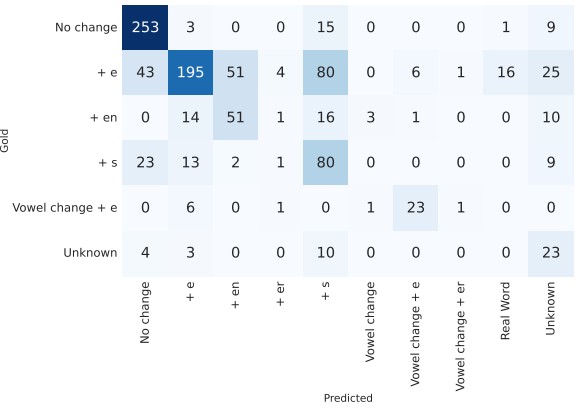

Figure 4: Confusion matrix for competing German plural morphemes for the one-shot setting.

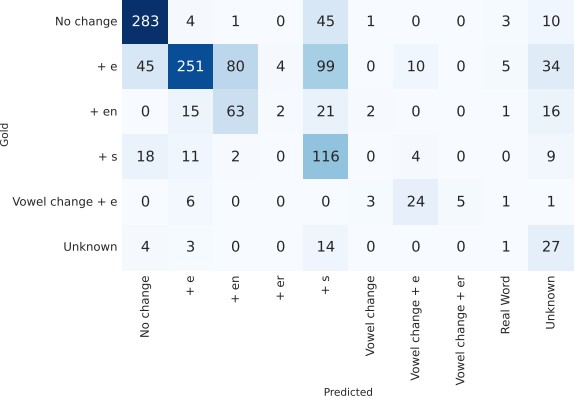

Figure 5: Confusion matrix for competing German plural morphemes for the zero-shot setting.

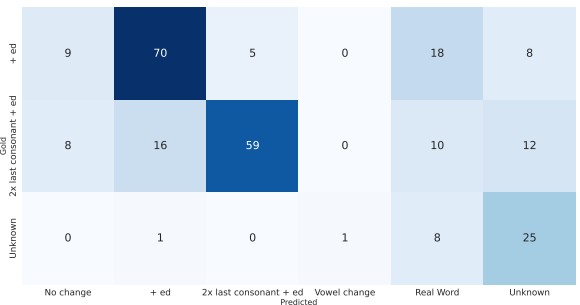

Figure 6: Confusion matrix for competing English past tense morphemes for the one-shot setting.

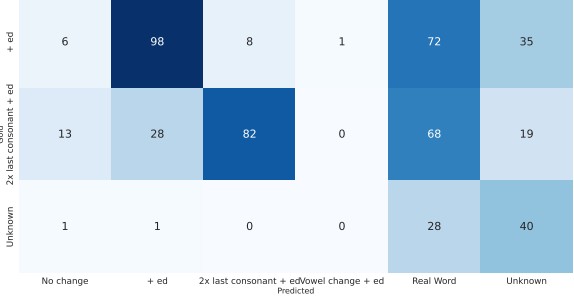

Figure 7: Confusion matrix for competing English past tense morphemes for the zero-shot setting.

design of intricate online surveys.[6]

We initiated the survey by presenting an introduction that detailed the concept of a wug test and the associated information for the survey. This introductory passage served to inform participants of the nature and intent of the research study, and it also provided examples to further facilitate their understanding of our task requirements.

Our data collection phase consisted of two parts: the English wug test and the German wug test. Upon consenting to participate, respondents were guided through a series of thoughtfully designed prompts related to the wug test. These prompts encouraged them to provide suitable responses based on their understanding of the task.

For the English wug test, we employed the following exemplary prompt: "Fill in the blank with the correct past tense of the word 'wug'. There is no predetermined correct answer. We encourage you to rely on your linguistic intuition. If you believe there are multiple possible responses, simply note the form that seems most accurate to you. For instance, 'They wug all the time. In fact, they __ just yesterday!'". Such prompts stimulated the participants to produce responses that were entirely their own, drawing on the provided information. For the German wug test, we translated the task instructions and prompts into German, ensuring easy comprehension for native German speakers.

In total, the English wug test incorporated 50 unique words for participants to respond to, while the German version consisted of 174 unique words. We received 28 responses for the English wug test and 21 responses for the German wug test.

### A.4 Other Values of $k$

Table 8 presents results for $k = 1$ and $k = 3$. Results for $k = 5$ are given in Section 5.

### A.5 Prompts

We leveraged the following prompts for the individual languages:

- English:
  - Long: "Fill in the blank with the correct past tense of the verb X. Answer with one word. They X all the time. In fact, they _ just yesterday! _ :"
  - Short: "Form the correct past tense of the verb X. Answer with one word. X :"

- German:
  - Long: "Fülle die Lücke mit dem korrekten Plural des Nomens X aus. Antworte mit einem Wort. Hier ist ein X. Jetzt sind es zwei _! _:"
  - Short: "Bilde den korrekten Plural des Nomens X. Antworte mit einem Wort. X :"

- Tamil:
  - Long: "நேற்று அவரிடம், "நீ X" என்றேன். அதைக் கேட்டு அவன் போய் _. _:"
  - Short: "X :"

- Turkish:
  - Long: "Boşlukları X ile verilen eylemin birinci tekil şahıs geçmiş zaman formları ile doldurun. Ben her zaman X. Ama dün _. _:"
  - Short: "Tek bir sözcük ile farazi X eyleminin birinci tekil şahıs geçmiş zaman hali nasıl olur? X :"

[6] https://www.qualtrics.com/

| Method | English | | German | | Tamil | |
|---|---|---|---|---|---|---|
| | $k = 1$ | $k = 3$ | $k = 1$ | $k = 3$ | $k = 1$ | $k = 3$ |
| ARL | 66.00 | 98.00 | 71.84 | 91.95 | 49.18 | 61.48 |
| MinGen | 56.00 | 60.00 | 39.66 | 60.92 | 39.34 | 49.18 |
| FIT | 84.00 ± 2.97 | 96.20 ± 0.60 | 70.06 ± 1.67 | 90.69 ± 0.99 | 44.75 ± 2.01 | 59.84 ± 3.07 |
| PPI 1 | 72.60 ± 6.00 | 90.80 ± 4.49 | 60.17 ± 6.80 | 82.59 ± 6.56 | 37.30 ± 2.57 | 51.07 ± 1.56 |
| AED | 44.20 ± 7.18 | 56.20 ± 6.54 | 27.82 ± 3.94 | 42.87 ± 4.65 | 46.15 ± 4.18 | 57.87 ± 5.46 |
| long 0-shot | 42.60 ± 4.90 | 55.60 ± 5.99 | 62.18 ± 2.45 | 81.55 ± 1.77 | 0.00 | 0.00 |
| long 1-shot | 58.40 ± 7.20 | 72.40 ± 6.50 | 63.36 ± 4.01 | 81.61 ± 3.09 | 5.27 ± 2.83 | 12.65 ± 6.19 |
| long few-shot | 57.60 ± 6.97 | 74.60 ± 4.90 | 65.86 ± 3.03 | 82.59 ± 1.96 | 15.25 ± 2.98 | 38.03 ± 4.18 |
| short 0-shot | 55.40 ± 6.07 | 72.20 ± 6.35 | 66.38 ± 2.48 | 84.43 ± 2.63 | 0.00 | 0.00 |
| short 1-shot | 61.20 ± 8.16 | 81.60 ± 6.97 | 67.31 ± 3.92 | 84.41 ± 2.36 | 1.99 ± 2.47 | 3.04 ± 3.58 |
| short few-shot | 61.40 ± 3.69 | 77.20 ± 3.12 | 68.97 ± 2.02 | 84.43 ± 1.07 | 17.05 ± 2.64 | 38.77 ± 2.86 |

Table 8: Results for other values of $k$.