# OpenReview forum: "Counting the Bugs in ChatGPT's Wugs: A Multilingual Investigation into the Morphological Capabilities of a Large Language Model"
_EMNLP/2023/Conference — EMNLP 2023 Main_

### Official Review · Reviewer_hkP1 · 2023-08-01

**Soundness:** 3

**Excitement:**

3: Ambivalent: It has merits (e.g., it reports state-of-the-art results, the idea is nice), but there are key weaknesses (e.g., it describes incremental work), and it can significantly benefit from another round of revision. However, I won't object to accepting it if my co-reviewers champion it.

**Paper Topic And Main Contributions:**

The article probes chatgpt on morphological skills by getting it to inflect non-words in English, German and Turkish amd Tamil. Based on the findings the chatgpt's internal representation of morphology seems to be lacking in what is generally being advertised. The article gives good linguistic backgrounds and hypotheses of morphological inflection relevant to the task and an experimental setup. The selection of languages is limtied and all belong in relatively well resourced languages with large speaker base and ample written material in the publically available data sets.

**Questions For The Authors:**

A> I am wondering of real world use cases of non-word morphology like this, I think if llm's could be harnessed as a tool to help e.g. linguists in lexicography and other work of e.g. lesser resourced languages, are there other directions to go from here besides probing potential capabilities or of llms?

**Reasons To Accept:**

* Article presents some evidence on linguistic capabilities of the internal representations of large language models, which is both very relevant to the venue and surprisingly rare topic

**Reasons To Reject:**

* there is a limited scientific value in merely using one off-the-shelf large language model via web-based api; i.e. reproducibility, issues etc.

**Reproducibility:**

1: Could not reproduce the results here no matter how hard they tried.

**Reviewer Confidence:**

4: Quite sure. I tried to check the important points carefully. It's unlikely, though conceivable, that I missed something that should affect my ratings.

---

> ### Author Rebuttal · Authors · 2023-08-29
>
> We would like to thank the reviewer for their helpful comments. Question A raises an important point. LLMs in general (and especially open source LLMs, as they improve in performance) encode a tremendous richness of information about human languages that can be tapped for analysis and can be used for multiple purposes:
> 1. To construct sets of stimuli for conducting quantitative studies and psycholinguistic experiments.
> 2. To perform experiments regarding grammatical patterns (including morphological patterns) that are latent within unlabelled language data (which is directly relevant to the current study).
> 3. To develop varied and challenging language teaching materials, especially for low-resource languages.

---

### Official Review · Reviewer_hyZR · 2023-08-03

**Soundness:** 4

**Excitement:**

3: Ambivalent: It has merits (e.g., it reports state-of-the-art results, the idea is nice), but there are key weaknesses (e.g., it describes incremental work), and it can significantly benefit from another round of revision. However, I won't object to accepting it if my co-reviewers champion it.

**Paper Topic And Main Contributions:**

The paper presents a study of the morphological capabilities of ChatGPT on the basis of the "Wug Test": The LLM is queried to provide an inflected or a derived form of a nonce word. The performance of the LLM and several baselines models is evaluated against human annotations. The performance of the baseline models is additionally evaluated on the SIGMORPHON 23 task. The "Wug" datasets are specially constructed for this study for four languages, German, English, Tamil and Turkish and will be published.

**Questions For The Authors:**

Question A: Did you use the openai provided version or the one on Microsoft Azure?
Question B: Beyond the investigation of the real-word bias in 6.4., did you find any other similarly groupable types of errors?

**Reasons To Accept:**

* Regarding large language models, "understanding their limitations, [...], is an ethical imperative" (l 654). I agree with that. The presented analysis of morphological capabilities of ChatGPT (first one, also to my knowledge) is a decent contribution to this respect.
* The study is broad, taking into account different morphological phenomena across different languages any many baseline models.
* Interesting findings regarding ChatGPT and morphology (e.g., l 555, l 605).
* The datasets are potentially valuable to further research.

**Reasons To Reject:**

The limitations section addresses the most important points already:
* Generalizability: Most importantly, the generalizability of the study beyond the combination of the particular GPT 3.5 version with the particular nonce word datasets is not clear.
* Tamil data: Futhermore, the inter-annotator agreement on the Tamil dataset is low, which might indicate a problem with the approach to its construction, also, the dataset is small.



**Reproducibility:**

3: Could reproduce the results with some difficulty. The settings of parameters are underspecified or subjectively determined; the training/evaluation data are not widely available.

**Reviewer Confidence:**

3: Pretty sure, but there's a chance I missed something. Although I have a good feel for this area in general, I did not carefully check the paper's details, e.g., the math, experimental design, or novelty.

**Typos Grammar Style And Presentation Improvements:**

* l 73, l 249: "LLMs" -> "ChatGPT".
* l 415: For the avoidance of any doubts regarding the evaluation, you could mention what "acc@k" stands for and why it makes sense as an evaluation metric.
* l 304: SIGMOPHON -> SIGMORPHON

---

> ### Author Rebuttal · Authors · 2023-08-29
>
> ### Generalizability: Most importantly, the generalizability of the study beyond the combination of the particular GPT 3.5 version with the particular nonce word datasets is not clear.
>
> We would like to point out that at the time of writing, there were no other large models available that were comparable in performance with GPT3.5 on morphological generalisation, or indeed reached the level at which an evaluation such as ours can be expected to return reasonable results. For instance, we found in initial experiments with Llama and Alpaca that there was no substantial morphological generalisation from nonce words. We had planned to include results from these models, but changed our plans when we discovered that they were not even remotely competitive with the GPT models, no matter how we constructed the prompts. This situation has since changed, with the release of open models such as Llama 2 (which actually does quite well at this task), and we heartily agree with the reviewer that a broader investigation is desirable for future work.
>
> ### Tamil data: Futhermore, the inter-annotator agreement on the Tamil dataset is low, which might indicate a problem with the approach to its construction, also, the dataset is small.
>
> We thank the reviewer for pointing out that our discussion of the low inter-annotator agreement (IAA) for Tamil is insufficient. We would like to draw the reviewer’s attention to two crucial points:
>
> Recent research in linguistics has shown that inter-speaker variability is more wide-spread in morphological systems than previously thought (see, for instance, Dammel and Schallert, 2019). Relatively low IAA for Tamil indicates that the examined morphological construction exhibits a particularly pronounced degree of variation — this is not a problem of our analysis, but an inherent property of morphology. For Tamil, the verb classes are not conditioned on phonemic or morphomeic features, “the correspondence between phonemic shape and class membership is not one that enables us to predict absolutely the latter from the former… in fact, pairs that are phonemically identical are found in different classes” (Lisker 1951). Because variation is grounded in historical factors, new verbs may belong to one of several classes based on how the speaker chooses to extend their knowledge of Tamil’s grammar, leading to low IAA. In other words, it is to be expected that IAA for Tamil annotators will be fairly low. We hope that the Tamil data (which we will release) might be an interesting starting point for future research into morphological variation in non-Indo-European languages.
> While not specifically focussing on morphological variation, our analysis is specifically set up such that it takes variation between humans into account: we evaluate using accuracy@k, i.e., if there is a high degree of variation between speakers, the model’s response is evaluated as correct if it is in line with any of the speakers (assuming there are not more than k answers in total, which is never the case for Tamil). This reflects our belief that when there is no one right answer for humans, we should not require this of the model either, but rather expect it to represent the full spectrum of variation in human judgement. Please note that this is exactly in line with a recent call in NLP to pay greater attention to annotator variation (e.g., Röttger et al., 2022).
>
> That being said, we now see that we should have devoted more attention to these points, and we will add a discussion to the final version of the paper.
>
> ### References
>
> Antje Dammel and Oliver Schallert. 2019. Morphological Variation: Theoretical and empirical perspectives. Amsterdam: John Benjamins.
>
> Paul Rottger, Bertie Vidgen, Dirk Hovy, and Janet Pierrehumbert. 2022. Two Contrasting Data Annotation Paradigms for Subjective NLP Tasks. In Proceedings of the 2022 Conference of the North American Chapter of the Association for Computational Linguistics: Human Language Technologies, pages 175–190, Seattle, United States. Association for Computational Linguistics.
>
> Lisker, Leigh. 1951. Tamil Verb Classification. Journal of the American Oriental Society 71.2, pages 111-114.

---

### Official Review · Reviewer_oAs1 · 2023-08-04

**Soundness:** 4

**Excitement:**

3: Ambivalent: It has merits (e.g., it reports state-of-the-art results, the idea is nice), but there are key weaknesses (e.g., it describes incremental work), and it can significantly benefit from another round of revision. However, I won't object to accepting it if my co-reviewers champion it.

**Paper Topic And Main Contributions:**

This paper examines the morphological 'understanding' of LLMs by prompting the production of nonce words. The authors focus on specific parts of inflection for English, German, Tamil and Turkish. The study promises a contribution of the datasets used. The results show that ChatGPT has yet to achieve human-level performance on the wug test.  Interestingly, the authors report 'real word biases' where a similar stem with the same grammatical features is produced (e.g., blushed for the past tense of blus).

**Questions For The Authors:**

(a) The final section refers to productivity, I wonder if the authors have considered a tangential thread: rare inflections of a paradigm? Presumably, some inflectional features will be more prevalent than others. For example, the future perfect continuous in the passive voice in English vs. Past tense.

(b) During the production stage of nonce words, the authors describe the process as :
[For English]: "These items were each perturbed by one or two letters (substituting phonetically similar sounds) producing a word not included"
[For Tamil]:  "combining syllables from real verb roots and checking against a Tamil dictionary"
[For German]: "We generated 200 nonce words with a length between four and seven characters"...using Unipseudo
[For Turkish]: The process isn't transparent/explained.
Is it possible that certain sound changes, or sequences can identify the word as a loan-word and therefore guide it's inflection pattern? How might letter frequency in each respective language affect your results (if at all)?

**Reasons To Accept:**

It's great to see linguistically motivated probing into LLMs. Given the recent hype, this kind of work is important. The authors present a set of experiments to test the morphological generalising capacities of LLMs - specifically gpt-3.5-turbo-0613.

**Reasons To Reject:**

I think claiming "typological diversity" is vague. My first thought in reading this description followed by English and German is that these languages are in the same language family! The authors should be specific, the research question narrows down the considered design space to morphology. So what morphological properties make these languages of interest?

I would've welcomed further discussion on the inter-annotator agreement. It was briefly mentioned in the Tamil results section. Why was inter-annotator agreement so low?

It feels as though the authors had a lot of results to share, which came at a cost of the narrative and explanation. There were a few conclusions that felt like a leap. For example, "factors other than inherent complexity play a role in ChatGPT’s ability to generalize morphological patterns". This feels like a bit of a throw-away line. Again, being more specific will help. Do the authors mean complexity in assigning morphological classes?

**Reproducibility:**

3: Could reproduce the results with some difficulty. The settings of parameters are underspecified or subjectively determined; the training/evaluation data are not widely available.

**Reviewer Confidence:**

4: Quite sure. I tried to check the important points carefully. It's unlikely, though conceivable, that I missed something that should affect my ratings.

**Typos Grammar Style And Presentation Improvements:**

-line193 what "it" refers to is unclear
-line596 I'm assuming the reference should be to figure 3 not 8.
-There were a few figures and tables which weren't referenced in text, so it took extra effort to see how they tied into the research presented.

---

> ### Author Rebuttal · Authors · 2023-08-29
>
> ### “I think claiming "typological diversity" is vague.”
>
> We thank the reviewer for encouraging us to provide more details about the specific ways in which the considered languages are typologically diverse. German and English were chosen since they exhibit morphological phenomena that have been investigated in great depth in the literature, thus providing a good opportunity for benchmarking LLMs. While English and German belong to the same language family (as rightly mentioned by the reviewer), the morphological system of German is considerably more complex and (in terms of typology) more fusional than English. Turkish was chosen since it is a non-Indo-European language with a fully agglutinative morphology, thus occupying a different point in the typological spectrum. Tamil was chosen since it is a Dravidian language exhibiting an agglutinative morphology with fusional elements. Thus, in terms of the classical triangle of fusional, isolating, and agglutinative morphologies, our chosen languages cover four different points: almost fully isolating (English), intermediate between isolating and fusional (German), intermediate between fusional and agglutinative (Tamil) and fully agglutinative (Turkish). Furthermore, the chosen languages also cover different points in the spectrum from low-resource to high-resource, enabling us to form hypotheses about the impact of the amount of language-specific training data on the morphological capabilities of LLMs.
>
> We agree that we should have discussed these points in greater detail in the paper and will do so in the final version.
>
> ### “I would've welcomed further discussion on the inter-annotator agreement.”
>
> We thank the reviewer for pointing out that our discussion of the low inter-annotator agreement (IAA) for Tamil is insufficient. We would like to draw the reviewer’s attention to two crucial points:
>
> Recent research in linguistics has shown that inter-speaker variability is more wide-spread in morphological systems than previously thought (see, for instance, Dammel and Schallert, 2019). Relatively low IAA for Tamil indicates that the examined morphological construction exhibits a particularly pronounced degree of variation — this is not a problem of our analysis, but an inherent property of morphology. For Tamil, the verb classes are not conditioned on phonemic or morphomeic features, “the correspondence between phonemic shape and class membership is not one that enables us to predict absolutely the latter from the former… in fact, pairs that are phonemically identical are found in different classes” (Lisker 1951). Because variation is grounded in historical factors, new verbs may belong to one of several classes based on how the speaker chooses to extend their knowledge of Tamil’s grammar, leading to low IAA. In other words, it is to be expected that IAA for Tamil annotators will be fairly low. We hope that the Tamil data (which we will release) might be an interesting starting point for future research into morphological variation in non-Indo-European languages.
>
> While not specifically focussing on morphological variation, our analysis is specifically set up such that it takes variation between humans into account: we evaluate using accuracy@k, i.e., if there is a high degree of variation between speakers, the model’s response is evaluated as correct if it is in line with any of the speakers (assuming there are not more than k answers in total, which is never the case for Tamil). This reflects our belief that when there is no one right answer for humans, we should not require this of the model either, but rather expect it to represent the full spectrum of variation in human judgement. Please note that this is exactly in line with a recent call in NLP to pay greater attention to annotator variation (e.g., Röttger et al., 2022).
>
> That being said, we now see that we should have devoted more attention to these points, and we will add a discussion to the final version of the paper.
>
> ### For example, "factors other than inherent complexity play a role in ChatGPT’s ability to generalize morphological patterns". This feels like a bit of a throw-away line.
>
> We agree that this was expressed clumsily. Our intent was to account for the high performance of ChatGPT on the German, relative to the English, wug test. One hypothesis might be that the English task was harder. However, in actual fact, the model could achieve a perfect score on the English dataset simply by guessing an -ed form for every verb, whereas the situation with German nouns was *much* more complicated (there was more variation and a richer set of conditions drove it). We speculated that ChatGPT does relatively poorly in English, then, not because the task is harder but because it has seen *too much* English and is suffering due to “real word bias.” The idea that we were trying to express was actually quite important and we would welcome the opportunity to express it better.
>
> ### “I wonder if the authors have considered a tangential thread: rare inflections of a paradigm?”
>
> We did consider this, but felt that tackling it was too much for this study. All of the inflectional categories explored here (English present versus past tense, Tamil present versus past tense, German singular versus plural, Turkish nominative versus accusative) are exceptionally common. Since high frequency inflectional categories tend to show greater variability (more “irregularity”) we focused our efforts there.
>
> ### “[For Turkish]: The process isn't transparent/explained.”
>
> We agree that the process of producing and annotating the Turkish data got short shrift. In brief, the data were generated by one author (an expert in the subject matter). They manually produced nonce roots consistent with the phonotactic rules of Turkish and checked to ensure that they did not correspond to dictionary words. We originally intended to recruit a team of Turkish annotators. However, this proved unnecessary since we discovered that the morphotactics and morphophonology of Turkish are so regular that the inflected forms followed deterministically from the form of the roots. For this reason, the annotations were produced by a single individual (unlike the other languages). Given an additional page, we will explain this matter (as well as the Tamil dataset)  in more detail.
>
> ###
> “Is it possible that certain sound changes, or sequences can identify the word as a loan-word and therefore guide its inflection pattern? How might letter frequency in each respective language affect your results (if at all)?”
>
> This is an excellent question (and one with which morphologists have wrestled for some time): are language users relying on formal factors (like sequences of sounds that make a new word sound “foreign”) when they decide what inflectional class to assign it to? Are they relying on meaning? Are they relying on explicit knowledge of the etymological origin of a word? Our study shows that German and Tamil speakers, at least, rely to a considerable extent on form. What we have not yet studied is whether the frequencies of graphemes in the stem (or, for that matter, the entropy of stems) predicts inflectional class (and whether LLMs like ChatGPT show the same behaviour). We think that this is an excellent avenue for future research.
>
> ### Typos Grammar Style And Presentation Improvements
>
> Thanks so much for drawing our attention to these mistakes — we will correct them in the final version of the paper!
>
> Antje Dammel and Oliver Schallert. 2019. Morphological Variation: Theoretical and empirical perspectives. Amsterdam: John Benjamins.
>
> Paul Rottger, Bertie Vidgen, Dirk Hovy, and Janet Pierrehumbert. 2022. Two Contrasting Data Annotation Paradigms for Subjective NLP Tasks. In Proceedings of the 2022 Conference of the North American Chapter of the Association for Computational Linguistics: Human Language Technologies, pages 175–190, Seattle, United States. Association for Computational Linguistics.
>
> Lisker, Leigh. 1951. Tamil Verb Classification. Journal of the American Oriental Society 71.2, pages 111-114.

---

### Meta-Review · Area_Chair_JWcj · 2023-09-11

**Recommendation:** 4

**Metareview:**

This paper investigated ChatGPT's morphological capabilities by prompting the generation of inflected or derived forms of nonce words. The findings indicate that ChatGPT has not yet reached human-level performance. Additionally, the creation of the dataset covering four languages is a contribution to the research community.

Testing only a single closed model through a web API is undeniably disappointing. However, as the authors aptly pointed out in their rebuttal, there was understandably no alternative available. I believe this paper deserves publication in some form.

---

### Decision · Program_Chairs · 2023-10-07

**Decision:**

Accept-Main

**Comment:**

This paper investigated ChatGPT's morphological capabilities by prompting the generation of inflected or derived forms of nonce words. The findings indicate that ChatGPT has not yet reached human-level performance. Additionally, the creation of the dataset covering four languages is a contribution to the research community.

Testing only a single closed model through a web API is undeniably disappointing. However, as the authors aptly pointed out in their rebuttal, there was understandably no alternative available. I believe this paper deserves publication in some form.